

# The use of genetic programming to develop a predictor of swash excursion on sandy beaches

Marinella Passarella[1], Evan B. Goldstein[2], Sandro De Muro[1], Giovanni Coco[3]

[1]Department of Chemical and Geological Sciences, Coastal and Marine Geomorphology Group (CMGG), Università degli Studi di Cagliari, 09124 Cagliari, Italy
[2]Department of Geological Sciences, University of North Carolina, 27599, U.S.A.
[3]School of Environment, Faculty of Science, University of Auckland, Auckland, 1142, New Zealand

*Correspondence to*: Marinella Passarella (marinella.passarella@unica.it)

## Abstract

We use Genetic Programming (GP), a type of Machine Learning (ML) approach, to predict the total and infragravity swash excursion using previously published datasets that have been used extensively in swash prediction studies. Three previously published works with a range of new conditions are added to this dataset to extend the range of measured swash conditions. Using this newly compiled dataset we demonstrate that a ML approach can reduce the prediction errors compared to well-established parameterizations and therefore it contributes to the error in coastal hazards assessment (e.g. coastal inundation). Predictors obtained using GP can also be physically sound and replicate the functionality and dependencies of previous published formulas. Overall, we show that ML techniques are capable of both improving predictability (compared to classical regression approaches) and providing physical insight into coastal processes.

## 1. Introduction

Wave runup, is the final expression of waves travelling from deep to shallow water and is directly associated to coastal hazards like flooding or erosion. Wave runup height can be defined from water level elevation time series at the shoreline $\eta(t)$, as the sum of two distinguished components: the wave set up (the temporal mean of the time series $\langle \eta \rangle$ relative to the still water level) and the swash $\eta'(t)$ (the vertical fluctuation of the water level around the wave set up). Understanding and predicting swash characteristics is critical for researchers seeking to understand the dynamics of fluid motions and sediment transport in the swash zone (e.g., Elfrink and Baldock, 2002; Masselink and Puleo, 2006), and for managers and practitioners addressing hazard setbacks, risk and coastal vulnerability (e.g., Bosom and Jimenez, 2011; Vousdoukas et al., 2012). Wave runup, (and therefore swash excursion) is a key component to evaluate inundation hazards and vulnerability to storm impacts (e.g., Bosom and Jiménez, 2011; Stockdon et al, 2007; Serafin et al., 2017). Stockdon et al., (2007) found that the wave action counted for about 48 % of the maximum total water level during two hurricanes along USA coast. The problem of





accurate predictions of wave runup and swash on sandy beaches has been a research topic for over 50 years but today we still struggle to provide reliable quantitative predictions.

The first predictors of wave runup were developed in the context of coastal structures (Miche, 1951; Hunt, 1959) and the formulas proposed, usually developed for steep slopes and under the assumption that runup motions reflect the standing component of the incident wave field. Overall these formulas suggested a dependence of the uprush elevation on

wave steepness and structure slope. A variety of predictors have since then been developed for vertical runup ($R$) and swash ($S$) on sandy beaches (e.g. Guza and Thornton, 1982; Holman and Sallenger 1985; Holman 1986; Ruessink et al., 1998, Stockdon et al., 2006), with details of the parameterizations depending on different combinations of deep water significant wave height ($H_{s0}$), deep water wave length ($L_0$) and beach slope ($\beta$). Guza and Thorton (1982) proposed a linear relationship between the significant runup ($R_s$) and $H_{s0}$ :

$$R_s = c\, H_{s0}\,,\tag{1}$$

where $c = 0.7$. Guza and Thornton (1982) also first distinguished between infragravity and incident swash components, indicating that the swash component related to low frequencies (infragravity, $R_{Ig}$) depends only on significant wave height (therefore excluding the beach slope) while the incident component can saturate as a result of the dissipative processes occurring in the surf zone. Their findings were later confirmed by several other studies although different dependencies on

environmental parameters were suggested (e.g. Holman and Sallenger, 1985; Ruessink et al., 1998).

Holman and Sallenger (1985) studying an intermediate to reflective beach (Duck, North Carolina, USA) described $R_s$ as:

$$R_s = c\, \xi_0\, H_{s0},\tag{2}$$

where $c$ is a constant, $H_{s0}/L_0$ is the wave steepness and

$$\xi_0 = \frac{\tan\beta}{\sqrt{H_{s0}/L_0}}\tag{3}$$

where $\beta$ is the foreshore beach slope and $\xi_0$ is the surf similarity index which is also often used for beach classification — beaches are classified as dissipative for values of $\xi_0 < 0.23$, reflective for $\xi_0 > 1$ and intermediate between the two (Short, 1999).

Stockdon et al., (2006) used 10 experiments from different locations to generate new parameterizations of wave

runup on natural beaches. The 2% exceedance value of wave runup $R_2$ was defined as:

$$R_2 = 1.1\left(\langle\eta\rangle + \frac{S_{Tot}}{2}\right),\tag{4}$$

where $\langle\eta\rangle$ is the maximum setup elevation and $S_{Tot}$ is the total swash defined as:



$$S_{Tot} = \sqrt{(S_{in})^2 + (S_{Ig})^2} \,, \tag{5}$$

where $S_{in}$ and $S_{ig}$ are the incident and infragravity components of swash. Stockdon et al. (2006) used regression techniques to

obtain relationships for $S_{in}$ and $S_{ig}$:

$$S_{in} = 0.75\beta\sqrt{H_0 L_0} \,, \tag{6}$$

and

$$S_{Ig} = 0.06\sqrt{H_0 L_0} \,. \tag{7}$$

Stockdon et al., (2006) is the most commonly used empirical parameterization of runup but, as can be noted

comparing eq. 6 and 7, the beach slope is missing from the predictor of the infragravity component of swash. The

dependency (or not) of $S_{ig}$ on beach slope is a topic that has been debated but not solved and some authors (e.g., Ruessink et

al., 1998) have indicated that infragravity swash is independent from the beach slope while a weak dependence on beach

slope has instead been reported by others (e.g., Ruggiero et al., 2004). Cohn and Ruggiero, (2016) suggested a bathymetric

control of the infragravity swash component through 1D and 2D numerical simulations performed using Xbeach (where

incident swash contribution is excluded) and compared them with previous formulas (Ruggiero et al., 2001; Stockdon et al.,

2006) and field data on dissipative beaches. They suggested that beach morphology (> -2 m MSL) influences the infragravity

component of runup more than the nearshore morphology (< -2 m MSL) and indicated that including the foreshore beach

slope in the formulation of $S_{Ig}$ improves predictability. Overall, it remains unclear if and when $S_{ig}$ depends on beach slope.

Finally, a number of other studies have also proposed other predictors that introduce other parameters to account for the

cross-shore wind component and the tidal range (Vousdoukas et al., 2012), the presence of nearshore sandbars (Cox et al.,

2013) or the sediment mean grain size for the case of gravel beaches (Poate et al., 2016). The above-mentioned empirical

runup formulas have been developed primarily with classic regression approaches (e.g.; Ruessink et al., 1998; Ruggiero et

al., 2001; Stockdon at. al., 2006; Vousdoukas et al., 2012).

Because of the importance of accurate predictions of swash excursion, the predictors provided by Stockdon et al.,

(2006) have been tested by various authors on beaches ranging from reflective to dissipative (e.g., Vousdoukas et al., 2012;

Cohn and Ruggiero, 2016; Atkinson et al., 2017). Predictions using Stockdon et al. (2006) are certainly sound (especially

considering the task of generating a universal formula for vertical swash excursion) even though differences between

measurements and predictions, possibly associated to local conditions, are inevitably found. More importantly, the regression

approach of multiple datasets first proposed by Stockdon et al. (2006) paves the way for our working hypothesis: can

powerful data-driven techniques be used to provide robust, reliable and realistic predictions of swash excursion?

When enough data exists, Machine Learning (ML) is a viable approach to regression problems. ML is a sub-

discipline of computer science focused on techniques that allow computers to find insightful relationships between variables

involved in swash processes, learning at each iteration (algorithm training and validation) from the provided dataset. A key



goal of ML is to develop predictors that are generalizable (able to describe the physical process beyond the training dataset

itself). Many different data-driven techniques fall under the purview of Machine Learning (e.g., decision trees, artificial neural networks, Bayesian networks, and evolutionary computation), all of which have shown applicability in coastal settings (e.g., Pape et al., 2007; Knaapen and Hulscher, 2002; Dickson and Perry, 2015; Yates and Le Cozannet, 2012). Previous Machine Learning work has focused on predicting runup and swash, but only for engineered structures, impermeable slopes, and/or for laboratory experiments (e.g., Kazeminezhad and Etemad-Shahidi, 2015; Bonakdar and

Etemad-Shahidi, 2011; Bakhtyar et al., 2008; Abolfathi et al., 2016) and not on natural beaches. In this study we focus on the use an evolutionary technique, Genetic Programming (GP), to solve the symbolic regression problem of developing new, optimized swash predictors.

In this contribution we first develop a swash excursion predictor using the original dataset of Stockdon et al., (2006), one of the most comprehensive studies in this area of research. In addition, we use data from Guedes et al., (2013),

Guedes et al., (2011, 2012), and Senechal, et al., (2011) to broaden the parameter space and to test the new swash equations. The data used in this work cover a broad range of swash excursion including extreme wave conditions (maximum $H_0 = 6.4$ m in Senechal, et al., 2011). High swash excursions, generated by extreme storms, are of particular interest when studying coastal hazards because they relate to flooding, beach and dune erosion (Bosom and Jimenez, 2011; Stockdon et al., 2007). The new ML derived results are also compared to the most widely used predictors from Stockdon et al., (2006). Finally, we

discuss the physical interpretation of the GP predictors and how we can use ML to gain knowledge of physical process related to the infragravity swash component.

## 2 Data

This work is based on two published video image-derived runup datasets — 13 field experiments in total. The first dataset (referred to here as the "original dataset") is composed by 491 swash measurements from 10 experiments aggregated by

Stockdon et al., (2006). The second dataset (referred to here as the "new dataset") consists of 145 swash measurements compiled for this work from three experiments performed by Guedes et al., (2013), Guedes et al., (2011), and Senechal, et al., (2011).

The compiled dataset of total swash is plotted in Fig. 1. The compilation of a large dataset deriving from 13 different experiments requires merging data collected using different techniques and equipment. Details of each experiment

can be found in the original references. Looking at the environmental forcing conditions, Figure 1 shows that the original and new dataset cover similar ranges of beach slope, while they differ in significant wave height (the new dataset includes wave heights over 6 m) and peak period (the original dataset includes more short period waves).



**Figure 1: Environmental forcing conditions (blue circles: original dataset, red crosses: new dataset): (a) significant wave height versus beach slope; (b) wave peak period versus beach slope; (c) significant wave height versus wave peak period.**

Both datasets include recordings of infragravity swash ($S_{Ig}$; m), total swash ($S_{Tot}$; m), beach slope ($\beta$) and concomitant offshore wave characteristics: significant wave height ($H_0$; m) and peak period ($T_p$; s). From these measurements the offshore significant wave length ($L_0$; m), wave steepness ($H_0/L_0$), and Iribarren number ($\xi_0$) were calculated. Experiments were located in North America, Europe and Oceania and cover a large range of the environmental condition (see Table 1 and Fig. 1).



**Table 1: Summary of wave and beach parameters for the original and the new datasets, beach name and type (following the classification of Short, (1999) based on Iribarren number (D stands for dissipative, I intermediate and R reflective); the last two rows indicates the range of parameters of the entire two datasets. Each experiment is associated to the citation where the measurements have been originally presented. If no reference is given, the citation to consider is Stockdon et al (2006).**


| Experiment | Dataset (data points) | Hs (m) | Tp (s) | $\beta$ | $\xi_0$ | Beach type | $S_{Tot}$(m) | $S_{Ig}$(m) |
|---|---|---|---|---|---|---|---|---|
| Duck 94 (Holland and Holman, 1996) | Original (52) | 0.7-4.1 | 3.8-14.8 | 0.06-0.1 | 0.33-1.43 | I, R | 0.8-2.9 | 0.5-2.2 |
| Gleneden | Original (42) | 1.8-2.2 | 10.5-16 | 0.03-0.11 | 0.26-1.2 | I, R | 1.1-2.3 | 0.9-1.9 |
| Sandy Duck | Original (95) | 0.4-3.6 | 3.7-15.4 | 0.05-0.14 | 0.34-3.22 | I,R | 0.7-2.3 | 0.3-1.8 |
| San Onofre | Original (59) | 0.5-1.1 | 13-17 | 0.07-0.13 | 1.6-2.62 | R | 0.9-2.6 | 0.5-1.8 |
| Terschelling (Ruessink et al., 1998) | Original (14) | 0.5-3.9 | 4.8-10.6 | 0.01-0.03 | 0.07-0.22 | D | 0.2-1 | 0.2-0.9 |
| Scripps Beach (Holland et al., 1995) | Original (41) | 0.5-0.8 | 10-10 | 0.03-0.06 | 0.4-0.92 | I | *0.3-0.7* | *0.3-0.7* |
| Delilah (Holland and Holman, 1993) | Original (138) | 0.5-2.5 | 4.7-14.8 | 0.03-0.14 | 0.44-1.70 | I,R | 0.7-3.3 | 0.4-1.7 |
| Duck 82 (Holman, 1986) | Original (36) | 0.5-4.1 | 6.3-16.5 | 0.09-0.16 | 0.68-2.38 | I,R | 0.7-3 | 0.4-2.4 |
| Agate (Ruggiero et al., 2001) | Original (14) | 1.8-3.1 | 7.1-14.3 | 0.01-0.02 | 0.1-0.19 | D | 0.7-1.5 | 0.7-1.5 |
| Ngarunui (Guedes et al., 2013) | New (32) | 0.6-1.26 | 8.1-12.4 | 0.01-0.03 | 0.13-0.42 | D | 0.24-0.9 | 0.24-0.9 |
| Tairua (Guedes et al., 2011) | New (25) | 0.7-1 | 9.9-12.5 | 0.09-0.13 | 1.4-2.25 | R | 1.2-2.2 | 0.6-0.95 |
| TrucVert (Senechal, et al., 2011). | New (88) | 1.1-6.4 | 11.2-16.4 | 0.05-0.08 | 0.49-0.9 | I | 0.81-2.5 | 0.63-2.37 |
| All beaches | Entire Original (Stockdon et al., 2006), (491) | 0.4-4.1 | 3.7-17 | 0.01-0.16 | 0.07-3.22 | D,I,R | 0.2-3.3 | 0.2-2.4 |
| All beaches | Entire New (145) | 0.6-6.4 | 8.1-16.4 | 0.01-0.13 | 0.13-2.25 | D,I,R | 0.24-2.5 | 0.24-2.37 |



Both datasets include all beach types, from dissipative to reflective. The two datasets also have a similar range of $S_{Tot}$ (although the original dataset records a larger swash, 0.2-3.3 m vs. 0.24-2.5 m of the new dataset), $S_{Ig}$ (about 0.2-2.4 m for

both), and $\beta$ (about 0.01-0.1 for both). The two datasets differ in the range of offshore wave conditions — in the original dataset $H_0$ and $T_p$ range over 0.4-4.1 (m) and 3.7-17 (s), respectively, while in the new dataset the ranges are 0.6-6.4 (m) and 8.1-16.4 (s).

The dissipative beaches of the original dataset (Fig. 2 d, h) are Terschelling and Agate, and for the new dataset Ngarunui (although, during the experiment, the beach also experienced intermediate conditions). The purely intermediate

beaches for the original and new dataset are Scripps and TrucVert. Some beaches of the original dataset represent both intermediate and reflective conditions: Duck 94, Gleneden, Sandy Duck, Delilah and Duck 82. San Onofre for the original and Tairua for new dataset are reflective.






**Figure 2 Total swash dependence on the environmental variables of the original (a,b,c,d) and new (e,f,g,h) datasets. The variables displayed are: significant wave height (a,e), wave peak period (b,f), beach slope (c,g) and Iribarren number (d,h). Beaches are considered dissipative (D) for values of ξ_0<0.23, reflective (R) for ξ_0>1 and intermediate (I) between the two (Short, 1999).**



## 3. Methodology

The large amount of data available (636 field swash records), including multidimensional variables, supports the feasibility of a ML approach. The data covers a wide range of environmental conditions (including extreme storms) and beach type, ensuring the applicability of our results to sandy beaches spreading from dissipative to reflective. We now outline the methods of the study. In Sect. 3.1, we present the supervised ML approach. In Sect. 3.2 we present the data pre-processing technique used to decide what data is shown to the ML algorithm. In Sect. 3.3 we discuss the techniques used to test the results from the ML algorithm against the testing data.

### 3.1 Genetic Programming

GP is a population-based machine learning approach based on evolutionary computation (Koza, 1992). The process of genetic programming can generally be divided into four steps: 1) an initial population of solutions for the problem is produced. For regression tasks such as developing a predictor for swash, the initial population of candidate solutions is in the form of equations (encoded as a tree or graph with a predefined mix of variables, operators and coefficients; Fig. 3). For step 2 of the routine the solutions are all compared to the training data to determine 'fitness' using a predefined error metrics; 3) the best solutions that minimise the error are proposed and the worst solutions are discarded; 4) new solutions created through 'evolutionary' rules (crossover via reproduction and mutation) and are added to the population of retained solutions. Steps 2 through 4 are repeated until the algorithm is stopped.

At the end of a routine, when the solutions have stabilized, a final population of solutions exist. A range of final solutions is given by the algorithm — more mathematically complex solutions (with more variables, operators, and coefficients) that minimize error are given alongside more simple, parsimonious solutions with higher error. These solutions exist along a "Pareto Front" that balances decreases in error with increasing solution complexity. Given a range of solutions with different error and complexity, we do not know of a perfect method for a user to determine the single best solution from the suite of final solutions — a user must decide on the solution according to different criteria: minimization of the error, computational time, physical meaning. In our work we adopted the criteria of minimization of the error with an eye toward the ability to interpret physical meaning from the formulas. A compromise between error reduction (more complex predictors) and ability of the predictors to generalize (predictive power on new data) should be found during the selection of a predictor.





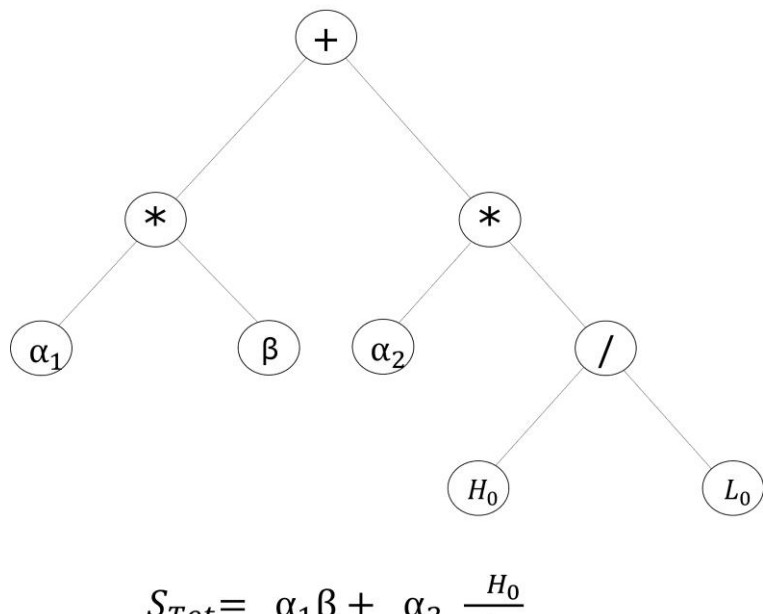

$$S_{Tot} = \alpha_1\beta + \alpha_2 \frac{H_0}{L_0}$$

**Figure 3: Schematics of the GP structure and principle of operation for an example of simple total swash predictor. $\alpha_1$ and $\alpha_2$ are the coefficients, $\beta$, $H_0$ and $L_0$ the variables and +, * and / the mathematical operators.**

All genetic programming in this study is performed using the software "Eureqa" developed by Schmidt and Lipson, (2009; 2013) which has successfully been used for a range of coastal problems (e.g., Goldstein et al., 2013; Tinoco et al., 2015). We searched for predictors of total and infragravity swash elevation — ultimately searching for the best equation that satisfies $S_{Tot/Ig} = f(H_0, L_0, \beta, T_p)$. Note also that we perform some experiments searching for total and infragravity swash as a function of composite variables like wave steepness, wave power ($P_w$), and the Iribarren number. However, the predictors

did not show improvement — also keep in mind that the GP can autonomously find these interrelationships between the basic parameters themselves, leading to the appearance of these composite variables in each optimization experiment. In addition to physical parameters, constants are included in the research and the mathematical operations allowed to the GP are: addition (+), subtraction (-), multiplication (*), division (/), exponential (^) and square root (√). Predictors developed on the training subset are assessed on the validation subset, using an error metric (also known as fitness function). From the

available metrics we selected the mean squared error (MSE):

$$MSE = \frac{1}{N}\sum_{i=1}^{N}(y_i - f(x)_i)^2 , \qquad\qquad\qquad (8)$$

where $N$ is the number of samples, $y_i$ is the measured value, and $f(x)_i$ is the value predicted by GP as a function of $x$. The search is stopped after that $10^{11}$ formulas were created and evaluated by the GP process, because no significant improvement in formula performance was found.



All selected formulas from the genetic programming routine are further optimized. First, the formulas are rearranged algebraically to ease interpretation by the user. Second, two coefficients of each selected formula are further optimized using a gradient descent algorithm in an iterative process.

**3.2 Training, Validation and Testing**

In order to obtain generalizable predictors, it is necessary to train, validate and test any ML routine on distinct and non-
overlapping subsets of data (e.g., Domingos, 2012). There is no universal, optimal method to select enough data to explain variability of the dataset while still retaining the most data to use for testing — recent work by Galelli et al., (2014) highlights that, even with the numerous input variable selection methods that have been proposed, there is no single best method for all the typologies of environmental datasets, and for all environmental models.

    We adopt the maximum dissimilarity algorithm (MDA) as selection routine (e.g., Camus et al., 2011), already
successfully tested in other works of predictors developed by GP for physical problems (e.g., Goldstein and Coco, 2014). The MDA is a routine for the selection of the most dissimilar points in a given dataset. Each data point is a vector composed by all the variables of our data set ($\eta, S_{Tot}, S_{Ig}, S_{in}, H_0, L_0, \beta, T_p, \xi_0, P_w, R$), where each variable is normalized between 0 and 1. At each iteration (i=1…n), the MDA finds the most different data point from the data selected in all previous iterations. Consequently the MDA selects a diverse set of data from the original 491 data points used by Stockdon et al.,
(2006). The operator must set the number of data points selected — we apply the MDA to 150 data points (~30% of the original dataset). We also run the analysis using a subset of variables (not including the variables representing swash elevations) but no significant loss in prediction power of the algorithms developed by the machine learning algorithm was observed). The data selected by the MDA is used as the training subset and we used the remaining data (~70 % of the original dataset, not selected) as validation subset.
The predictors developed by the GP using this training data was tested using the new dataset from Guedes et al., (2011), Guedes et al., (2013), and (Senechal, et al., 2011). This new dataset is completely independent from the training and unknown at the GP algorithm providing a test in the ability of the GP parameterization to generalise, even beyond the range of the testing and validation data (Fig. 1). The performance of our predictors using the testing data is compared to the Stockdon et al., (2006) predictors using the error metrics in Sect. 3.3.


**3.3 Error Evaluation**

We use three different error metrics for the testing phase and for comparing our predictor with known predictors in the literature. The mean square error as defined in Eq. (8), the root mean square error:

$$RMSE = \sqrt{\frac{1}{N}\sum_{i=1}^{N}(y_i - f(x)_i)^2},$$  (9)





and the maximum absolute error:

$$MaxAE = max_{i=1\ldots N}(y_i - f(x)_i) \,, \tag{10}$$

where $N$ is the number of samples, $y_i$ is the measured value, and $f(x)_i$ is the value predicted by the GP as a function of $x$.

## 4 Results

### 4.1 Results of GP experiments

After $\sim10^{11}$ formulas were evaluated, the solutions from the GP algorithm for both $S_{Tot}$ and $S_{Ig}$ follow a "Pareto front" where the error decreases (compared with the validation subset) as the size (or complexity) of the formula increases. Several viable techniques exist for selecting the best solution to avoid overfitting, all meant to balance the fact that simpler solutions (the minimum description length) might risk losing more accurate information contained in more complex models (e.g., O`Neill et al., 2010). Generally, extremely complicated predictors fit the training and validation dataset better than simpler

predictors but they may lose generalization power (overfitting). Picking a solution is a subjective task, and relies on specific domain knowledge on the part of the user — here we focus on predictors with clear physical plausibility (avoiding predictors with physical nonsense such as increase of $S_{Tot}$ as $H_0$ decreases) and avoid predictors that are difficult to interpret, (e.g., extremely nonlinear relationships, possibly a result of overfitting the training dataset). We also focus on two predictors for both the $S_{Tot}$ and $S_{Ig}$, evaluating a simpler and more complex predictor to determine if the more complex expression

warrants use when generalized to the testing dataset.

### 4.2 Total Swash

Following the principle of error reduction and physical interpretability of the results, we finally selected from the pool of candidate solutions available from the GP experiments, two formulas for $S_{Tot}$, one simpler (Eq. 11) and one more elaborated (Eq. 12).

$$S_{Tot} = 12.314\,\beta + 0.087\,T_p - 0.047\,\frac{T_p}{H_0} \,, \tag{11}$$

$$S_{Tot} = 146.737\,\beta^2 + \frac{T_p H_0{}^3}{5.800 + 10.595\,H_0{}^3} - 4397.838\,\beta^4 \,. \tag{12}$$

Note that the coefficients of both Eq. (11) and (12) are dimensional. Eq. (11) represents the best solution in terms of error reduction while maintaining a physical interpretability. It also stands out for its simplicity and only weak nonlinearity — it looks similar to a multiple linear regression. In Eq. 12 the first and the third term depend exclusively on $\beta$, while the

second term includes the contribution of the incident waves. The total swash in both GP predictors is related to the wave peak period (instead of wave length) different from previous formulations (e.g. Stockdon et al., 2006; Holman and Sallenger 1985). Recently also Poate et al., (2016) used the wave peak period in their runup predictor for gravel beaches. The use of





the peak period instead of the wave length has no influence on the physics of the predictor, but could allow the users a more direct utilization of the formula.

Figure 4 displays a comparison of performance of swash predictors obtained through the ML approach (Fig. 4a, 4b) and Stockdon et al. (2006) (Fig 4c), on the training and validation dataset. This does not constitute a test of the predictors, only a consistency check to see that the predictors are modelling the training and validation data appropriately. Overall Eq. (5) shows a higher scatter in the whole original dataset (details on the errors can be found in Table 2 and Sect. 4.4). It is not clear why all formulas do not successfully fit the data Duck 82 and Delilah (especially for $S_{Tot} > 2$ m). The Stockdon et al.,

(2006) predictor shows scatter at larger total swash, while the GP predictors shows slight under fitting of swash elevation during large events. Stockdon et al. (2006), Eq. (5), (6) and (7) in this contribution, mostly under predicts the data with exclusion of the Duck 82 dataset, which is largely over predicted for high values of the swash excursion. Both GP predictors more accurately fit data from dissipative beaches (Agate and Terschelling) compared with the Stockdon et al. (2006) formula.


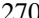

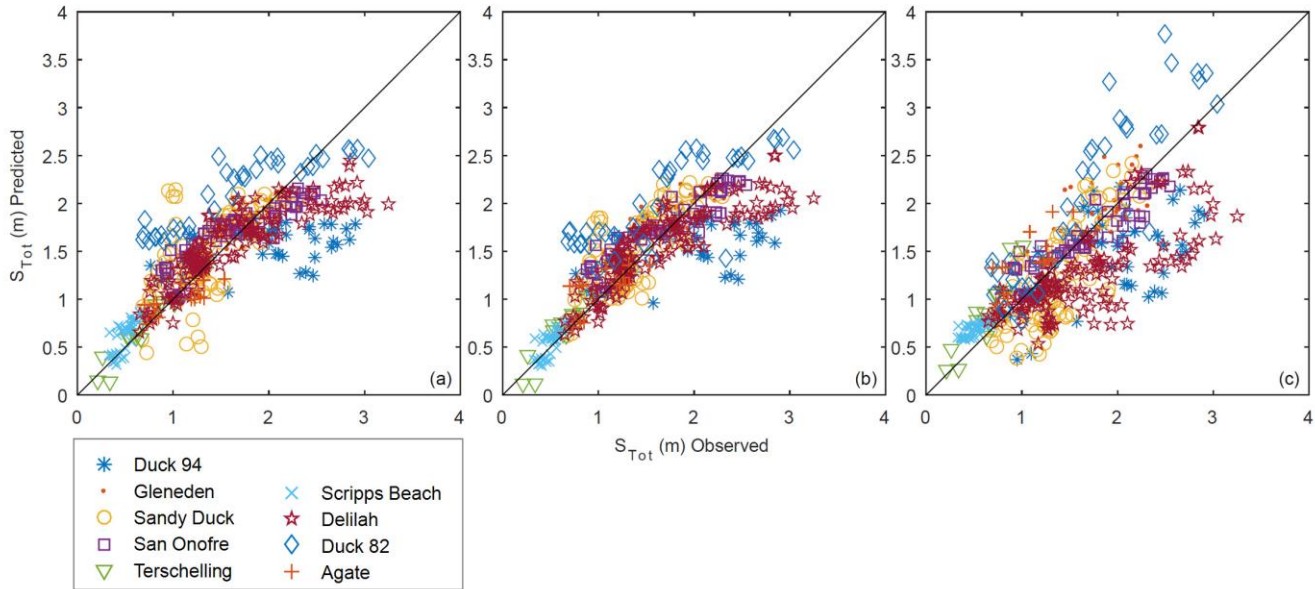

**Figure 4: Observed versus predicted $S_{Tot}$ using (a) GP Eq. (11), (b) GP Eq. (12) and(c) Stockdon et al., (2006) Eq. (5) for the original dataset (Stockdon et al., 2006). This is not a test of any predictor, only a consistency check — all data was shown to the GP**
**algorithm and used to generate the linear regression in panel c.**

Figure 5 shows the observed versus the predicted $S_{Tot}$ — both GP models and Stockdon et al., (2006) — for the new, 'testing' dataset. Note that swash values (0- 2.5 m) are lower than the maxima observed in the original data set, but these values represent absolutely new, out of sample prediction for all equations. Overall the Stockdon et al., (2006) formula has higher scatter than both GP predictors (Fig. 5), and considerably overestimates swash measurement of Truc Vert



(intermediate beach under extreme highly energetic wave storm) and Ngarunui (dissipative beach under mild wave conditions) while underestimates the observations at the reflective beach of Tairua. Equations (11) and (12), from the GP routine, perform similarly (Fig. 5 a, b).

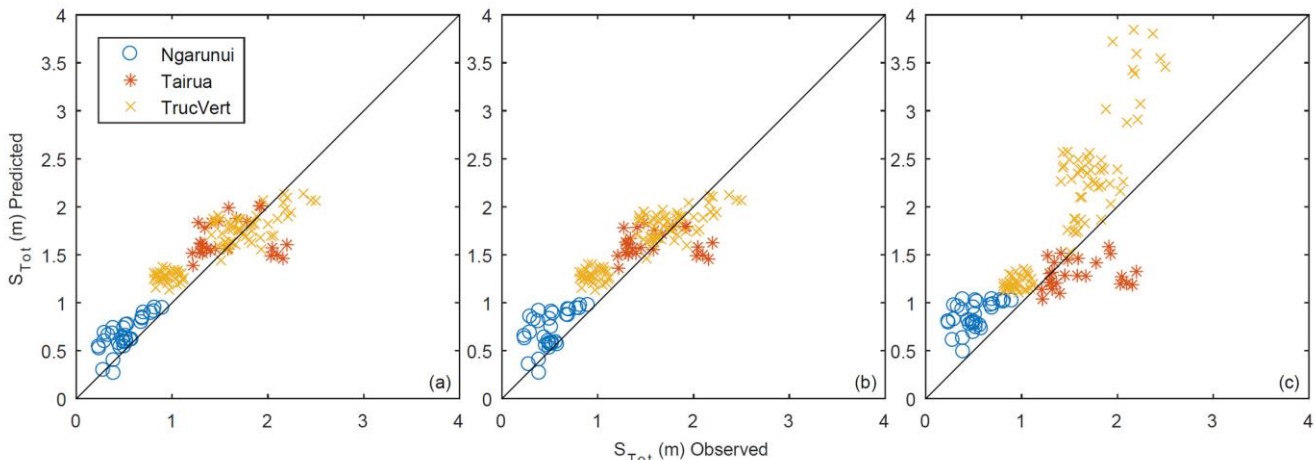

**Figure 5: Observed versus predicted $S_{Tot}$ with the new independent dataset. (a) GP Eq. (11) , (b) GP Eq. (12) and (c) Stockdon et al., (2006) — Eq. (5) in this manuscript.**

### 4.3 Infragravity swash

The two formulas selected for describing $S_{Ig}$ are Eq. (13) and the more complex Eq. (14):

$$S_{Ig} = 10\beta + \frac{\beta}{\beta - 0.306} + \frac{H_0 - 0.456}{0.447 + 136.411(\frac{H_0}{L_0})} \tag{13}$$

$$S_{Ig} = \frac{\beta}{0.028 + \beta} + \frac{(-1)}{2412.255\,\beta - 5.521\,\beta L_0} + \frac{H_0 - 0.711}{0.465 + 173.470(\frac{H_0}{L_0})}, \tag{14}$$

As in the case of $S_{Tot}$, the coefficients of Eq. (13) and (14) for $S_{Ig}$ are dimensional. The reader should also note that both formulas depend on the beach slope in contrast with Ruessink et al., (1998) and Stockdon et al., (2006), Eq. (7) in this manuscript, but in agreement with other slope inclusive predictors (Ruggiero et al., 2001; 2004). Eq. (14) represents the best solution in terms of error reduction while maintaining physical meaning and Eq. (13) is a simpler predictor where the

contribution of beach slope and waves to infragravity swash remains separate. Both Eq. (13) and (14) have the same nonlinear term $\frac{H_0 - 0.456}{0.447 + 136.411(\frac{H_0}{L_0})}$ , with slight difference in the coefficients, that describes the incoming waves. The threshold that flips this term from negative to positive is related to wave height and is probably an indication that for small waves the infragravity component is extremely limited (this terms needs to be negative to compensate for other terms that only depend on beach slope and provide a constant contribution). The ML predictor not only suggests that the beach slope is important




when predicting infragravity swash, but also indicates a nonlinear interaction between waves and beach morphology through the wave length (second term of Eq. 14).

Figure 6 displays a consistency check, highlighting the performance of swash predictors obtained through a ML approach (Fig 6a, 6b) and Stockdon et al., (2006) (Fig 6c), on the training and validation dataset. It is not clear why all formulas provide less precise prediction with data from Duck 84 and Duck 82 but we note that these two experiments focused on

intermediate to reflective conditions with relatively large wave conditions (Table 1). Generally the three formulas seem to perform similarly. Some differences are found in the overestimation of Agate and Terschelling data from Stockdon et al., (2006), while the GP predictors show less scatter.

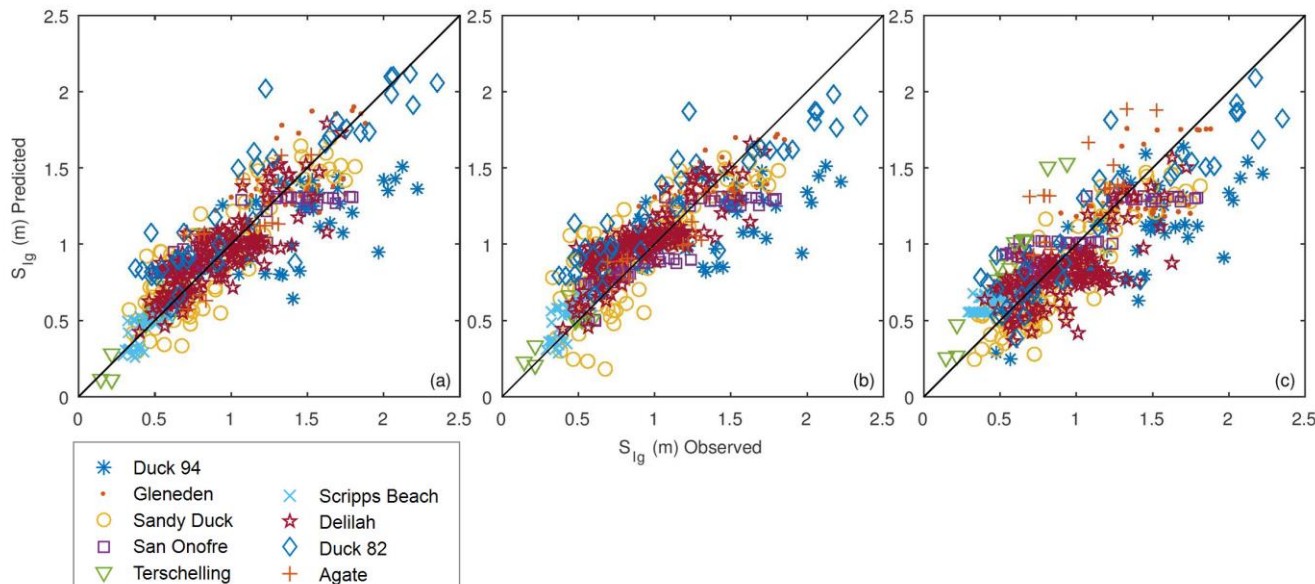

**Figure 6: Observed versus predicted $S_{Ig}$ by (a) GP Eq. (13), (b) GP Eq. (14) and (c) Stockdon et al., (2006) Eq. (7) on the original dataset (Stockdon et al., 2006). This is not a test of any predictor, only a consistency check — all data was shown to the GP algorithm and is the same data used to generate the linear regression in panel c.**

The same difficulty in predicting swash excursion on a dissipative beach is observed on Ngarunui (Fig. 7). Note that this experiment was performed under mild wave conditions ($H_0 \sim$ 0.6-1.26 (m) and $T_p \sim$8.1-12.4 (s), Table 1) compared to the

experiments at, Agate and Terschelling. Also Truc Vert presents dissipative conditions in the swash zone, while the surf zone is intermediate ($\xi_0$ up to 0.87 as reported by Senechal et al., 2011). For this experiment Eq. (13) and (7) (Fig. 7 a, c) overestimate $S_{Ig}$ while Eq. (14) performs clearly better, suggesting that it could be the most appropriate for $S_{Ig}$ predictions.

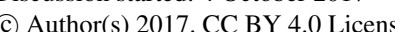



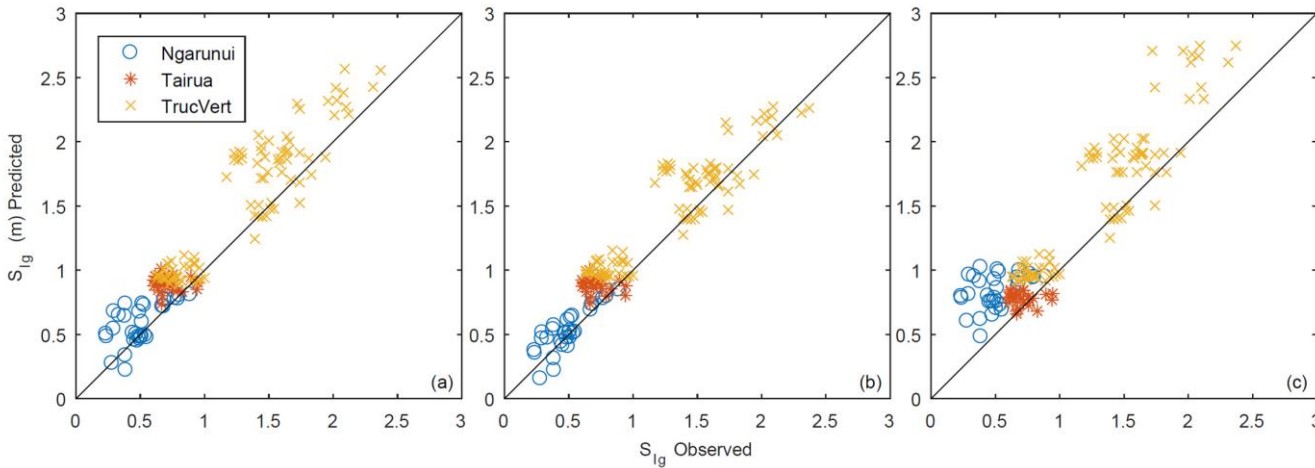

**Figure 7: Observed versus predicted $S_{Ig}$ using (a) GP Eq. (13), (b) GP Eq. (14) and (c) Stockdon et al., (2006) Eq. (7) on the new independent dataset.**

Table 2 summarises the results of the errors calculated, through three error metrics (Sect. 3.3), of the two GP predictors and the Stockdon et al., (2006) formulas on both original and independent datasets.





**Table 2: Results of error metrics for both total and infragravity swash, calculated for the GP predictors and Stockdon et al., (2006) on both original and new datasets. The results calculated with the original dataset (in Italics) do not represent a test of any predictor, only a consistency check — all original data was shown to the GP algorithm and used by Stockdon et al., (2006).**

| Target | Formula (Methodology) | Error Metrics | New Dataset (Independent) | Original Dataset Stockdon et al., (2006) |
|---|---|---|---|---|
| Total Swash | Eq. (11) (GP) | MSE (m$^2$) | 0.074 | *0.144* |
| | | RMSE (m) | 0.272 | *0.380* |
| | | MaxAE (m) | 0.695 | *1.257* |
| | Eq. (12) (GP) | MSE (m$^2$) | 0.083 | *0.126* |
| | | RMSE (m) | 0.288 | *0.355* |
| | | MaxAE (m) | 0.702 | *1.258* |
| | Eq. (5) Stockdon et al., (2006) | MSE (m$^2$) | 0.325 | *0.214* |
| | | RMSE (m) | 0.570 | *0.462* |
| | | MaxAE (m) | 1.771 | *1.399* |
| Infragravity | Eq. (13) (GP) | MSE (m$^2$) | 0.071 | *0.047* |
| | | RMSE (m) | 0.267 | *0.217* |
| | | MaxAE (m) | 0.679 | *1.019* |
| | Eq. (14) (GP) | MSE (m$^2$) | 0.047 | *0.053* |
| | | RMSE (m) | 0.216 | *0.231* |
| | | MaxAE (m) | 0.587 | *1.025* |
| | Eq. (7) Stockdon et al., (2006) | MSE (m$^2$) | 0.111 | *0.068* |
| | | RMSE (m) | 0.334 | *0.261* |
| | | MaxAE (m) | 0.988 | *1.056* |

Overall the GP predictors perform better than the Stockdon et al., (2006) formulation for all the error metrics considered and for the new testing datasets (for both $S_{Tot}$ and $S_{Ig}$). While for $S_{Tot}$ the predictor of smaller size performs better than the more complex predictor, for $S_{Ig}$ the errors decrease with increasing GP predictor size (Eq. (13) to (14)), when tested on the new dataset. Eq. (11) has the smallest RMSE (0.272 m), MSE (0.074 m$^2$) and MaxAE (0.695 m) of the $S_{Tot}$ formulas, evaluated on the new dataset, while the predictor from Stockdon et al. (2006) — Eq. (5) of this manuscript — has the highest RMSE

(0.570 m), MSE (0.325 m$^2$) and MaxAE (1.771 m). Eq. (14) performs slightly better than Eq. (13) in predicting $S_{Ig}$ evaluated on the new dataset, while the difference is larger when compared to the predictor from Stockdon et al. (2006) — Eq. (7) of this manuscript.




## 5 Discussion

In this work we use data compiled by Stockdon et al., (2006) to build new predictors, by the use of GP, for both total and
infragravity swash elevations. We then test the generalizability of these new predictors using new data (including some extreme conditions). This is different from the usual use of a single dataset, divided into three parts for training, validation and testing of ML derived predictors. We did not assume a single criteria for the selection of the best predictors, but we found a compromise between error reduction (on the testing dataset) and the physical interpretability of the results.

Results demonstrate that the GP predictors proposed in this work perform better than existing formulas and that ML
can identify nonlinear relationships between the variables of this problem. Specifically, Eq. (14) introduces the dependence of $S_{Ig}$ on the beach slope, but also its nonlinear relationship with the wave length. Furthermore, solutions for $S_{Ig}$ found by the GP algorithm with the smaller size (not shown) show a simple linear dependence on $H_0$ with a constant (identical to early formulation of wave runup e.g. of Guza and Thornton, 1982). More complex predictors add a dependence on $L_0$, $\sqrt{H_0 L_0}$ (similar to Eq. 7 of this manuscript — from Stockdon et al., 2006) and $\frac{H_0}{L_0}$.

The GP algorithm found solutions for $S_{Ig}$ that include the beach slope ($\beta$), a variable that is never excluded from predictors of further increasing size. Because the candidate solutions resulted from GP experiments follow a "Pareto front" distribution in which the increase in fitting (smaller MSE) grows as the size of the formula rises, the continuous inclusion of $\beta$ for more complex predictors implies that including $\beta$ in $S_{Ig}$ formulation reduces prediction error. The improvement of classic empirical techniques, by innovation in data-driven methodologies, has already been discussed (e.g. the case of depth-
averaged velocities over model vegetation by Tinoco et al., 2015). Experiments based on GP also highlighted a way to focus on and add dependencies in predictors describing coastal processes (e.g. grain size in the case of prediction of ripple wave length by Goldstein et al., 2013). The predictors proposed in this work perform well on a wide range on environmental conditions, including, as defined by Nicolae et al., (2016), the highest stormy condition dataset (Truc Vert beach) recorded in the field and available in the literature. Furthermore the work here demonstrates that ML derived results, when physically
plausible, may be generalizable beyond the limits of the training data, extrapolating to a novel, out of sample data set.

Looking at the limitation of the proposed models, the variables taken into account ($H_0$, $T_p$, $L_0$, $\beta$) are easily accessible but also oversimplify the processes that affect swash. For instance, we do not include the influence of the wave directional spread (Guza and Feddersen, 2012), the cross-shore wind component and the tidal range (Vousdoukas et al., 2012). However, in order to include these and other aspects (e.g., role of underwater vegetation) it is necessary to perform
more field experiments that record swash, runup and other relevant variables. An additional limitation is that the swash formulas obtained in this study approaches a nonzero value as wave height approaches zero. While this is physically incorrect, the data used in the analysis does not include the limit condition of 'no waves-no swash'. Consequently, even if the GP formulas obtained do not correctly predict the limit condition corresponding to a no wave scenario, the prediction for both datasets has smaller errors compared to commonly used formulas.



Our results contribute to the discussion on the role of beach slope on the prediction of the infragravity component of swash. The GP algorithm found an $S_{Ig}$ dependence on beach slope and increasingly more complicated formulas (i.e., more precise predictions) found by the GP all include beach slope as one of the predictive variables. This result is in line with studies such as Ruggiero et al., (2001 and 2004) and in contrast with Stockdon et al., (2006), Senechal et al. (2011) and Ruessink et al., (1998).

Our results are relevant for a variety of applications where the errors related to empirical formulation obtained by classic regression techniques could be reduced. For instance in the case of coastal hazards, Stockdon et al., (2006) formulation for wave runup is used by Serafin and Ruggiero, (2014) for their extreme total water level estimation and by Bosom and Jimenez, (2011) in their framework for coastal hazards assessment. Accuracy in runup formulation has consequences for risk and vulnerability assessment as coastal management maps (De Muro et al. in press; Perini et al.,

2016), and other several studies regarding sediment transport (Puleo et al., 2000), swash zone hydrodynamics and morphodynamics (Puleo and Torres-Freyermut, 2016).

## 6 Conclusions

        Starting form a large dataset covering a wide range of swash, beach and wave field characteristics, we developed two new predictors for total and infragravity swash elevations, using the machine learning technique of Genetic

Programming. We tested and compared our new formulas with previously developed and largely accepted parameterizations of swash (e.g., Stockdon et al., 2006) using independent published datasets. Results of the two GP predictors selected (one for total and one for infragravity swash) show better performance compared with the formulation of Stockdon et al., (2006), evaluated using an independent (unknown to the algorithm) dataset (which included extreme highly energetic wave storm, particularly relevant for coastal hazards). This work contributes to reducing the uncertainty in predicting the swash excursion

and consequently in assessing the coastal vulnerability and hazards (e.g. inundation) which depend in part upon wave swash (Bosom and Jimenez, 2011). A better prediction of swash excursion could also influence retreat or accommodation strategies and integrated planning for the mitigation of coastal hazards. Furthermore, GP results indicate that the beach slope influences the infragravity component of the swash — GP predictors improve in performance when the beach slope was included. We therefore conclude that beach slope is a relevant parameter when predicting the infragravity component of the

swash elevation, even though this is contrary to several previous studies (e.g., Stockdon et al., 2006; Ruessink 1998; Senechal et al., 2011). ML and specifically GP can be a useful tool for data-rich problems providing robust predictors and possibly also physical insight. The role and importance of the scientist is not reduced or substituted by the machine but instead improved thanks to a powerful data analysis tool.





**Competing interests**

The authors declare that they have no conflict of interest.

**Acknowledgments**

MP gratefully acknowledges Sardinia Regional Government for the financial support of her PhD scholarship (P.O.R. Sardegna F.S.E. Operational Programme of the Autonomous Region of Sardinia, European Social Fund 2007-2013 - Axis IV
Human Resources, Objective l.3, Line of Activity l.3.1.)". MP gratefully acknowledges the support and funding GLOBUSDOC international placement programme of University of Cagliari. MP and SD gratefully acknowledge the support and funding of by the Project N.E.P.T.U.N.E. (Natural Erosion Prevision Through Use of Numerical Environment) L. R. 7.08.2007, N.7: ''Promozione della ricerca scientifica in Sardegna e dell`Innovazione tecnologica in Sardegna''. GC funded by a GNS-Hazard Platform grant (contract 3710440). EBG gratefully acknowledges the support of UoA through a
PBRF grant.

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
