# Peer review of "The use of genetic programming to develop a predictor of swash excursion on sandy beaches"

_Natural Hazards and Earth System Sciences, 2017_

## Referee Comment (RC1) · Anonymous Referee #1 · 26 Oct 2017

I have read the manuscript with pleasure and a lot of interest and I think it is an important contribution. The manuscript is also well written and the presentation of the results is good. Therefore I am very favorable to its publication in your journal I only have some minor comments/suggestions which I feel would improve the manuscript. The authors compiled a new wave run up dataset, by extending the already broad Stockdon et al dataset with other measurements. Following they apply machine learning to conclude that the latter performs better in predicting wave run up heights. Machine learning in general have been already shown to be very capable predictors of wave runup. One of the earliest examples can be found here (https://link.springer.com/article/10.1007/s10236-011-0440-5) and maybe the most re-

cent before the present work is Abolfathi et al 2016, with all studies reporting very good results. Since empirical run up formulas are simplifications of the actual processes deriving site specific 'recalibrations' of existing wave run up formulas has been considered a recommended practice. That way at least the effect of some unkown parameters is reduced. I would recommend the authors to mention that somewhere when they discuss previous studies, since among the recent ones mostly Stockdon 2006 aimed to propose a universal parameterization. Wave setup is part of run up and is driven by wave breaking. The latter is controlled by the nearshore beach slope (and not of the beachface only) a parameter which most times remains unknown, among others. Moreover, infragravity motions and wave setup are not the same thing but they could be confused in some field measurements. The authors could elaborate on these aspects when they discuss infragravity parameterizations. The weak point of machine learning techniques is that their predictive skill is limited to the conditions covered by the parameter space of the training dataset. GP is superior in that aspect to ANNs, since the final product is a relationship that is based on parameterizations which were derived considering the physical processes. At the same time it is not meant that the coefficients estimated will result in reasonable results beyond the range of the training dataset. In this case the training dataset is quite extensive but given that most of the global coastline is not included, it is not for granted that the solution could fail in other parts of the world. All this is not criticism, I just think the authors should discuss the above points. In addition I believe that it will be helpful for the reader to provide information about the range of input parameters for which the formulas are valid (maybe in form of a table).

---

## Referee Comment (RC2) · Anonymous Referee #2 · 1 Nov 2017

This is an interesting paper and an important contribution to the prediction of wave runup and swash excursion. Results are presented in a clear and focussed way. The manuscript is well written with a suitable and comprehensible outline. The figures are well readable and have informative captions. The basic idea and the application of the genetic programming algorithm are presented and discussed in a clear manner also understandable for scientists not familiar with machine learning techniques. The discussion of the physical meaning of the generated predictors is very valuable. A comparison with the results of previous predictors on the same dataset gives is well done and illuminates the advantages of the Genetic Programming approach. The manuscript should be published in the journal in its present state.

---

## Referee Comment (RC3) · Anonymous Referee #3 · 1 Nov 2017

This is a well written paper that proposes a new methodology to predict total and infra gravity swash elevation. As such it is of interest to NHESS and coastal scientists/practitioners. The methodology followed is correct and well explained. In particular, there is a very clear explanation of Genetic Programming and how this technique has been used for this work. This is very well written in a way which is suitable for non-experts approaching the methodology for the first time. The data used are of very good quality and there is a good explanation of the range of parameters covered by the dataset. The results are discussed in concise and detailed way and the accuracy improvement over existing relationships is demonstrated.

[Figure]

My only minor suggestion is that the use of both MSE and RMSE is redundant and one of the two can be omitted.

Therefore I recommend publication after minor revision.

Minor corrections/suggestions.

Abstract Line 14: change the sentence: it contributes to the error, maybe it is contributes to the reduction of the error. However, beware of repetitions.

Also many repetitions of "wave runup" in the introduction, try to rephrase.

Line 123, concomitant is possibly better replaceable by "associated".

Line 143-147: specify the countries of the beaches named as not all authors might be familiar with these.

Line 170. You might want to specify which is your stopping criterion, and when do you consider the solutions stable.

Line 237: overfitting is mentioned, but it could be useful to explain what this is in the present context. Explanation in 240 occurs after the first use of the term and it is not clear.

Some sentences are written in present tense (e.g. we use at the beginning of Section 3.3, and "...finally selected" at the start of 4.2). Please make the tense consistent.

Also, in Line 314 it is mentioned that experiments in Ngarunui beach are carried out under mild dissipative conditions. Is the difficulty in predicting these results due to the particular combination of H and T (hence L)? It would be useful to be more detailed in explaining this.

In Line 356 it is claimed that the procedure followed is different from the use of a single data set. This needs clarification, as you always build one dataset that is divided in three for training validation and testing. The same was done in the development of

ANN tools for overtopping in the CLASH project (van Gent et al. 2007), for example, when the dataset used was actually a composite one resulting from many datasets.

References

van Gent, M.R., van den Boogaard, H.F., Pozueta, B. and Medina, J.R., 2007. Neural network modelling of wave overtopping at coastal structures. Coastal Engineering, 54(8), pp.586-593.
* * *

---

## Author Comment (AC1) · 5 Dec 2017

**Response to Referee #1**

**Referee comments are in** plain text
**Author comments are in BOLD**
**NHESSD Manuscript text is in** *italics*
**Added Text is in** *Bold Italics*
**(Line numbers refer to the marked manuscript)**

**We thank the Referee for the comments and address each point below.**

I have read the manuscript with pleasure and a lot of interest and I think it is an important contribution. The manuscript is also well written and the presentation of the results is good. Therefore I am very favorable to its publication in your journal I only have some minor comments/suggestions which I feel would improve the manuscript. The authors compiled a new wave run up dataset, by extending the already broad Stockdon et al dataset with other measurements. Following they apply machine learning to conclude that the latter performs better in predicting wave run up heights. Machine learning in general have been already shown to be very capable predictors of wave runup. One of the earliest examples can be found here (https://link.springer.com/article/10.1007/s10236-011-0440-5) and maybe the most recent before the present work is Abolfathi et al 2016, with all studies reporting very good results.

**Thank you for pointing us toward the manuscript of Vousdoukas et al (2011). We now cite this work on Line 97-99:**

*"Previous Machine Learning work has focused on predicting runup and swash, but only for engineered structures, impermeable slopes, and/or for laboratory experiments (e.g., Kazeminezhad and Etemad-Shahidi, 2015; Bonakdar and Etemad-Shahidi, 2011; Bakhtyar et al., 2008; Abolfathi et al., 2016) and not on natural beaches **apart from Vousdoukas et al., (2011) which used artificial neural network (ANNs) for shoreline contour elevation (which includes the wave runup), on a natural beach in Portugal**."*

Since empirical run up formulas are simplifications of the actual processes deriving site specific 'recalibrations' of existing wave run up formulas has been considered a recommended practice. That way at least the effect of some unknown parameters is reduced. I would recommend the authors to mention that somewhere when they discuss previous studies, since among the recent ones mostly Stockdon 2006 aimed to propose a universal parameterization.

**This is a good point, and we now added it to the discussion section.**

**Line 404-406:**

*"Generally the results from machine learning technique are strictly related to the range of the training and validation datasets (original dataset in Fig. 1). This work demonstrated that the applicability of the predictors can sometimes be used beyond the range of the testing dataset (new dataset in Fig. 1). However it is unknown how predictors will perform in settings beyond those in the present work — future tests on new field data are therefore recommended. **Furthermore, parameterizations always work better when free parameters are optimized to a given site by using existing data and it should be considered when proposing universal parameterizations.**"*

Wave setup is part of run up and is driven by wave breaking. The latter is controlled by the nearshore beach slope (and not of the beachface only) a parameter which most times remains unknown, among others.

**We added a comment that nearshore slope, which is a controlling parameter, is excluded from this predictor**

**Line 393-397**

*"Looking at the limitation of the proposed models, the variables taken into account ($H_0$, $T_p$, $L_0$, $\beta$) are easily accessible but also oversimplify the processes that affect swash. For instance, we do not include the influence of the wave directional spread (Guza and Feddersen, 2012), the cross-shore wind component and the tidal range (Vousdoukas et al., 2012). However, in order to include these and other aspects (e.g., role of underwater vegetation,* **nearshore bathymetry***) it is necessary to perform more field experiments that record swash, runup and other relevant variables. An additional limitation is that the swash formulas obtained in this study approaches a nonzero value as wave height approaches zero."*

**and we clarify the concept at:**

**Line 411-413**

*"This result is in line with studies such as Ruggiero et al., (2001 and 2004) and in contrast with Stockdon et al., (2006), Senechal et al. (2011) and Ruessink et al., (1998).* ***Although difficult to quantify and extremely simplified (this parameter together with sediment diameter should integrate the effect of the entire cross-shore profile), our results suggests that some parameter involving the beach profile should be considered when predicting runup characteristics.***"

Moreover, infragravity motions and wave setup are not the same thing but they could be confused in some field measurements. The authors could elaborate on these aspects when they discuss infragravity parameterizations.

**We distinguishing wave setup from infragravity motions, we state at the beginning of the manuscript:**

**Line 2-5**

*"The height reached by waves can be defined from water level elevation time series at the shoreline η(t) as the sum of two distinguished components: the wave set up (the temporal mean of the time series ⟨η⟩ relative to the still water level) and the swash η′(t) (the vertical fluctuation of the water level around the wave set up)."*

**and we have added (on line 74-75):**

*"**In addition the similarity in the temporal scales of wave setup and infragravity motions could also be a confounding factor in measurements.** Finally, a number of other studies have also proposed other predictors that introduce other parameters to account for the cross-shore wind component and the tidal range "*

The weak point of machine learning techniques is that their predictive skill is limited to the conditions covered by the parameter space of the training dataset. GP is superior in that aspect to ANNs, since the final product is a relationship that is based on parameterizations which were derived considering the physical processes. At the same time it is not meant that the coefficients estimated will result in reasonable results beyond the range of the training dataset. In this case the training dataset is quite extensive but given that most of the global coastline is not included, it is not for granted that the solution could fail in other parts of the world.

**First, we would like to point out that this GP routine is not 'aware' of physical processes (some previous GP work by other researchers have been forced to conform to physical laws). Only parameters and coefficients to combine these variables were given to the GP (along with the data).**

**We have clarified some aspects of our work in the discussion, and how it relates to generalization and extrapolation beyond the range of input data.**

**Line 366-367:** "*In this work we use data compiled by Stockdon et al., (2006) to build new predictors, by the use of GP, for both total and infragravity swash elevations. We then test the generalizability of these new predictors using new data (including some extreme conditions). **This is different from previous applications of ML in coastal settings in two ways: First, we are testing the ML-derived predictor on data that is collected from a different setting (compared to the training data)— three beaches not included in the training data. Second, the testing data includes events that are outside the data range of the training data — we are extrapolating the ML-derived predictor as a test of its generalizability.***"

**Line 400-406**

"*... the data used in the analysis does not include the limit condition of 'no waves-no swash'. Consequently, even if the GP formulas obtained do not correctly predict the limit condition corresponding to a no wave scenario, the prediction for both datasets has smaller errors compared to commonly used formulas. **Generally the results from machine learning technique are strictly related to the range of the training and validation datasets (original dataset in Fig. 1). This work demonstrated that the applicability of the predictors can sometimes be used beyond the range of the testing dataset (new dataset in Fig. 1). However it is unknown how predictors will perform in settings beyond those in the present work — future tests on new field data are therefore recommended.***"

All this is not criticism, I just think the authors should discuss the above points. In addition I believe that it will be helpful for the reader to provide information about the range of input parameters for which the formulas are valid (maybe in form of a table).

**We have included the information regarding the range of parameters on which the formulas have been developed on Figure 1, where the blue circles represent the training and validation dataset and the red crosses the testing one. Figure 2 includes the entire range of input and target parameters for which these predictors have been evaluated and tested (and therefore for which their validity has been assessed in the present work).**

**Moreover details on parameters intervals divided for experiment are reported in Table 1. For highlighting the aspects you suggested we included the following sentences at line 401-407**

**Line 401-407**

*"... the data used in the analysis does not include the limit condition of 'no waves-no swash'. Consequently, even if the GP formulas obtained do not correctly predict the limit condition corresponding to a no wave scenario, the prediction for both datasets has smaller errors compared to commonly used formulas.* **Generally the results from machine learning technique are strictly related to the range of the training and validation datasets (original dataset in Fig. 1). This work demonstrated that the applicability of the predictors can sometimes be used beyond the range of the testing dataset (new dataset in Fig. 1). However it is unknown how predictors will perform in settings beyond those in the present work — future tests on new field data are therefore recommended."**

**Thank you for considering the revised version of this manuscript for publication in NHESSD**

---

## Author Comment (AC2) · 5 Dec 2017

**Response to Referee #2**

**Referee comments are in** plain text
**Author comments are in BOLD**

This is an interesting paper and an important contribution to the prediction of wave runup and swash excursion. Results are presented in a clear and focused way. The manuscript is well written with a suitable and comprehensible outline. The figures are well readable and have informative captions. The basic idea and the application of the genetic programming algorithm are presented and discussed in a clear manner also understandable for scientists not familiar with machine learning techniques. The discussion of the physical meaning of the generated predictors is very valuable. A comparison with the results of previous predictors on the same dataset gives is well done and illuminates the advantages of the Genetic Programming approach. The manuscript should be published in the journal in its present state.

**We thank the Referee for reading the manuscript and providing these positive comments.**

---

## Author Comment (AC3) · 5 Dec 2017

**Response to Referee #3**

**Referee comments are in** plain text
**Author comments are in BOLD**
**Manuscript text is in** *italics*
**Added Text is in** *Bold Italics*
**Line numbers refer to the marked manuscript**

**We thank the Referee for the comments and address each point below.**

This is a well written paper that proposes a new methodology to predict total and infragravity swash elevation. As such it is of interest to NHESS and coastal scientists/practitioners. The methodology followed is correct and well explained. In particular, there is a very clear explanation of Genetic Programming and how this technique has been used for this work. This is very well written in a way which is suitable for non-experts approaching the methodology for the first time. The data used are of very good quality and there is a good explanation of the range of parameters covered by the dataset. The results are discussed in concise and detailed way and the accuracy improvement over existing relationships is demonstrated.

My only minor suggestion is that the use of both MSE and RMSE is redundant and one of the two can be omitted. Therefore I recommend publication after minor revision.

**We prefer to keep both MSE and RMSE in the manuscript to aid in the rapid comparisons with future prediction schemes — for instance, a future study may report only MSE or RMSE.**

Minor corrections/suggestions.
Abstract Line 14: change the sentence: it contributes to the error, maybe it is contributes to the reduction of the error. However, beware of repetitions.

**We addressed this.**

**Line 14:** *"Using this newly compiled dataset we demonstrate that a ML approach can reduce the prediction errors compared to well-established parameterizations and therefore **it may improve** coastal hazards assessment (e.g. coastal inundation)."*

Also many repetitions of "wave runup" in the introduction, try to rephrase.

**Line 33**

**We changed** *"The first predictors of wave runup were…"* **into** *"The first predictors of **these phenomena** were…"*

Line 123, concomitant is possibly better replaceable by "associated".

**Line 126: We replaced** *"concomitant"* **with** *"associated"*

Line 143-147: specify the countries of the beaches named as not all authors might be familiar with these.

**We added the information about the countries where the experiments were performed.**

**Line146-150: "***The dissipative beaches of the original dataset (Fig. 2 d, h) are Terschelling* **(Netherlands)** *and Agate* **(USA)***, and for the new dataset Ngarunui* **in New Zealand** *(although, during the experiment, the beach also experienced intermediate conditions). The purely intermediate beaches for the original and new dataset are Scripps* **(USA)** *and TrucVert* **(France)***. Some beaches of the original dataset* **(USA)** *represent both intermediate and reflective conditions: Duck 94, Gleneden, Sandy Duck, Delilah and Duck 82. San Onofre for the original and Tairua* **(New Zealand)** *for new dataset are reflective.*"

Line 170. You might want to specify which is your stopping criterion, and when do you consider the solutions stable.

**We moved and clarified the sentence from lines 203 – 204 to line 173-175**

**Line 173-175: "The search is stopped after the GP evaluated $10^{11}$ formulas because the solutions stabilized and no significant improvement in formula performance was found."**

Line 237: overfitting is mentioned, but it could be useful to explain what this is in the present context. Explanation in 240 occurs after the first use of the term and it is not clear.

**We define overfitting before mentioning it (moved to line 242) and we add a definition of overfitting from a new reference: Dietterich T.: Overfitting and Undercomputing in Machine Learning, ACM Comput. Surv., 27 (3), doi:10.1145/ 212094.212114 1995.**

**Line 242-248:** *"Generally, extremely complicated predictors fit the training and validation dataset better than simpler predictors but they may lose generalization power* **when tested on a separate testing dataset (overfitting). In other words a predictor with overfitting could represent the noise in the training and validation subsets instead of defining a general predictive rule (e.g., Dietterich, 1995) and therefore it will result in smaller training errors but in higher testing errors.** *Several viable techniques exist for selecting the best solution to avoid overfitting, all meant to balance the fact that simpler solutions (the minimum description length) might risk losing more accurate information contained in more complex models (e.g., O`Neill et al., 2010)."*

Some sentences are written in present tense (e.g. we use at the beginning of Section 3.3, and "...finally selected" at the start of 4.2). Please make the tense consistent.

**We changed the past tense into present tense.**

**Line 127, 128**

**We changed "*were calculated*" to "*are calculated*" and "*were located*" to "*are located*"**

**Line 192**

**We changed** *"We searched"* **to "*We searched*"**

**Line 223, 225**

**We changed** *"we used"* **to "*we use*", We changed** *"was tested"* **to "*is tested*",**

**Line 256**

**We changed** *"selected"* **to "*select*"**

**Line 374-375**

**We changed** *"did not"* **to "*do not*", and** *"found"* **to "*find*"**

Also, in Line 314 it is mentioned that experiments in Ngarunui beach are carried out under mild dissipative conditions. Is the difficulty in predicting these results due to the particular combination of H and T (hence L)? It would be useful to be more detailed in explaining this.

**This sentence is connected to the previous one where we discussed that the Stockdon et al. formula for $S_{Ig}$ has more scatter for Terschelling and Agate (which are dissipative). We did not highlighted the characteristic of these beaches in the paragraph, but we already defined them dissipative at line 277 (because the same happen for $S_{tot}$). Our intention on line 323 was to highlight the performance of the predictors on the dissipative beaches — settings where infragravity motion has the greatest importance. Ewe now clarify this on line 314-329.**

**Line 314-329:** *"Generally the three formulas seem to perform similarly. Some differences are found in **dissipative settings (i.e.,** Agate and Terschelling) —**predictions by** Stockdon et al., (2006) **tend to overestimate $S_{ig}$ compared to** the GP predictors . The same difficulty in predicting swash excursion on a dissipative beach is observed on Ngarunui (Fig. 7). **Even though** this experiment was performed under mild wave conditions ($H\_0 \sim 0.6$-$1.26$ (m) and $T\_p \sim 8.1$-$12.4$ (s), Table 1) compared to the experiments at Agate and Terschelling. **Note that dissipative beaches are the one were the infragravity motion has greater importance**. Also Truc Vert presents dissipative conditions in the swash zone, while the surf zone is intermediate ($\xi\_0$ up to 0.87 as reported by Senechal et al., 2011). For this experiment Eq. (13) and (7) (Fig. 7 a, c) overestimate $S\_Ig$ while Eq. (14) **has better performance for the dissipative beach Ngarunui,** suggesting that it could be the most appropriate for $S\_Ig$ predictions."*

In Line 356 it is claimed that the procedure followed is different from the use of a single data set. This needs clarification, as you always build one dataset that is divided in three for training validation and testing. The same was done in the development of ANN tools for overtopping in the CLASH project (van Gent et al. 2007), for example, when the dataset used was actually a composite one resulting from many datasets.

References
van Gent, M.R., van den Boogaard, H.F., Pozueta, B. and Medina, J.R., 2007. Neural network modelling of wave overtopping at coastal structures. Coastal Engineering,

54(8), pp.586-593.

**We now clarify our study — discussing how our method is different than previous data splitting work.**

**Line 368-372: "***In this work we use data compiled by Stockdon et al., (2006) to build new predictors, by the use of GP, for both total and infragravity swash elevations. We then test the generalizability of these new predictors using new data (including some extreme conditions). This is different from* many previous applications of ML in coastal settings in two ways: First, we are testing the ML-derived predictor on data that is collected from a different setting (compared to the training data)— three beaches not included in the training data. Second, the testing data includes events that are outside the data range of the training data — we are extrapolating the ML-derived predictor as a test of its generalizability.**

**Thank you for considering the revised version of this manuscript for publication in NHESSD**